# Purification and Identification of Antioxidant Peptides from Rice Fermentation of *Lactobacillus plantarum* and Their Protective Effects on UVA−Induced Oxidative Stress in Skin

**DOI:** 10.3390/antiox11122333

**Published:** 2022-11-25

**Authors:** Qiuting Mo, Shiquan You, Hao Fu, Dongdong Wang, Jiachan Zhang, Changtao Wang, Meng Li

**Affiliations:** 1Beijing Key Laboratory of Plant Resource Research and Development, College of Chemistry and Materials Engineering, Beijing Technology and Business University, Beijing 100048, China; 2Institute of Cosmetic Regulatory Science, Beijing Technology and Business University, Beijing 100048, China

**Keywords:** *Lactobacillus plantarum*, fermentation, polypeptides, antioxidant

## Abstract

Oxidative stress is an important factor on both aging and disease. Among foods endowed with beneficial healthy properties, rice is a very useful material, not only because it has a good amino acid ratio and produces antioxidant peptides through microbial fermentation, but also for its inexpensive availability. In this study, rice was treated with *Lactobacillus plantarum*, and the resulting mixture of small peptides with less than 11 amino acids (RFP) was extracted and purified from the fermentation broth. Subsequently, the antioxidant activity of RFP was assessed using the chemical model, cell biology, and animal model methods. RFP enhanced the expression of the antioxidant enzyme genes downstream of the KEAP1−NRF2/ARE pathway by promoting nuclear factor−erythroid 2−related factor 2 (NRF2) nuclear translocation while simultaneously removing lipid oxidation products and excess free radicals. These results suggest that RFP is a potential substance for resisting aging and disease caused by oxidative stress.

## 1. Introduction

Oxidative stress refers to the imbalance of oxidative and antioxidant systems in the body under endogenous or exogenous stimulation, leading to the production of excessive free radicals, which results in damage to tissue cells and biological macromolecules and is associated with various diseases such as atherosclerosis, diabetes, cancer, and aging [1]. As the largest organ of the human body, the skin is the first barrier against external environmental stimulation, such as ultraviolet (UV) radiation, chemical smoke, and haze [2]. UV radiation is the main external cause of skin oxidative stress, accelerating aging and disease. UVA (320–400 nm) penetrates the epidermis of the skin to the dermis and is the strongest form of UV radiation on the Earth’s surface [3].

After the oxidative stress theory was put forward, research on antioxidants followed. As a natural antioxidant, antioxidant peptides have the advantages of a comparatively simple structure, ease of absorption, high stability, and a lack of immunogenicity, for which they are drawing increasing attention in food, medicine, and other fields [4]. Rice has a protein concentration of approximately 8% and such advantages as a good amino acid composition ratio and low sensitization, making it suitable for the production of antioxidant peptides [5]. Currently, enzymatic hydrolysis is generally used to produce antioxidant peptides, a method which has the benefits of high specificity and a reasonably mature technique, but high cost. Nevertheless, the preparation of bioactive peptides by microbial fermentation has the advantages of simplicity, low production cost, and easy industrialization. *Lactobacilli* have evolved an efficient proteolytic system that can degrade large proteins into small peptides and amino acids, making them an important group of bacteria involved in protein degradation and active peptide generation in the fermentation process [6].

The chemical model method, the cell biology method, and the animal model method are the three techniques used to assess the antioxidant activity of antioxidant peptides. Only a small percentage of investigations into the antioxidant activity of peptides are conducted using cell biology or animal models; the majority use chemical model approaches [7]. However, peptides with strong antioxidant activity at the biochemical level do not necessarily have the same effect on organisms. Therefore, it is necessary to combine biological tests for evaluation on the basis of chemical method evaluation. [8].

This study examined the antioxidant activity of the rice fermentation polypeptides (RFP) that were extracted from rice fermentation broth fermented by *Lactobacillus plantarum*. First, the chemical model determined the free radical scavenging and ferric reduction ability of RFP. Then, the effects of RFP on the scavenging ability of reactive oxygen species (ROS) and malondialdehyde (MDA), and on the overall antioxidant capacity of human skin fibroblasts (HSFs) induced by UVA oxidative stress were verified. The effects of RFP on the KEAP1−NRF2/ARE pathway were also confirmed. Finally, we preliminarily detected the effects of RFP on the catalase (CAT) enzyme and total antioxidant capacity in UVA−irradiated mice and observed the changes in mouse skin through tissue sections.

## 2. Materials and Methods

### 2.1. Materials

Rice was purchased from Wuchang City, Heilongjiang Province, China; *Lactobacillus plantarum* (Orla−Jensen) Bergey, Harrison, Breed, Hammer & Hu (Lactobacillaceae) was obtained from the China Center of Industrial Culture Collection (strain number: CICC−20261); rice polypeptides (Lys−His−Asn−Arg−Gly−Asp−Glu−Phe) (RP) were purchased from Top−peptide Biotechnology Co., Ltd. (Shanghai, China); Human Skin Fibroblasts (HSFs) were obtained from the Cell Bank of the Chinese Academy of Sciences; Cell Counting Kit−8 (CCK−8) was purchased from Biorigin (Beijing) Inc. (Beijing, China); Dulbecco’s Modified Eagle’s Medium (DMEM) was purchased from Invitrogen (Carlsbad, CA, USA); fetal bovine serum (FBS), trypsin−EDTA and penicillin−streptomycin were purchased from Gibco (Carlsbad, CA, USA); RIPA Lysis Buffer (P0013D), nuclear and cytoplasmic protein extraction kit, trizol, BCA protein assay kits, CAT, superoxide dismutase (SOD), glutathione peroxidase (GSH−Px), total antioxidant capacity, and MDA and ROS kits were purchased from Shanghai Beyotime Biotechnology (Shanghai, China); and all other chemicals employed were of high−purity biochemistry grade. EasyScript^®^ One−Step gDNA Removal and cDNA Synthesis SuperMix and TransStart^®^ Top Green qPCR SuperMix (Beijing, China) were also utilized. A Human Nuclear Factor E2−related Factor 2 (NRF2) ELISA Kit, Human Quinone NADH Dehydrogenase 1 (NQO1) ELISA Kit, and Human Heme Oxygenase−1 (HO−1) ELISA Kit were purchased from Cusabio Biotech Co., Ltd. (Houston, TX, USA). All chemicals and solvents used in the experiments were of analytical grade and purchased from Sinopharm Chemical Reagents Co., Ltd. (Shanghai, China).

### 2.2. Preparation of Rice Fermentation Broth

The MRS medium was configured and sterilized by autoclaving for 15 min. After cooling to room temperature, the *Lactobacillus plantarum* suspension was added for expansion culture. The culture conditions were 37 °C and 180 rpm for 24 h. The rice was dried in an oven at 60 °C to constant weight, then crushed using a grinder and filtered through a 50−mesh screen. Deionized water was added at a material−to−liquid ratio of 1:4. After sterilization, 5% activated *Lactobacillus plantarum* liquid (10^7^ CFU/mL) was added, and fermentation was performed at 180 r/min at 37 °C. After centrifugation, the polysaccharides of the supernatant were removed using the alcohol precipitation method and the ratio of ethanol to supernatant was 4:1. After freeze−drying at minus 60 °C in vacuum rotation, the freeze−dried rice fermentation liquid powder was obtained, and the yield was 0.17 g/L.

### 2.3. Purification and Identification of Antioxidant Peptides from Fermentation Broth

#### 2.3.1. Separation and Purification

An activated ion exchanger (SP Sepharose Fast Flow) [9] gel was wet−loaded (1.6 × 20 cm) and allowed to settle naturally, then balanced with deionized water for 12 h. The lyophilized powder was prepared into 4g/L solution with distilled water and filtered through a 0.22mm pore size polyether sulfone filter membrane to remove insoluble matter. The sample loading volume was 2 mL and the flow rate was 2 mL/min. They were then eluted sequentially with deionized water, 0.1 NaCl solution, and 1 mol/L NaCl solution at a flow rate of 2 mL/min. Thirty tubes of each eluent were collected, each containing about 10 mL. The protein concentration of the eluent was detected, and the elution curve was made with the same absorbance value as the ordinate and with the same elution volume as the abscissa. The same peak components were combined. Finally, the packing was cleaned with 2.0 mol/L NaCl at a flow rate of 2 mL/min to regenerate the ion exchange column.

#### 2.3.2. Molecular Weight Detection

The molecular weight of the polypeptides was determined by reversed−phase high−performance liquid chromatography (RP−HPLC) [10]. The determination was performed on a TSKgel 2000SWXL 300 × 7.8 mm column at 25 °C. The mobile phase was a 0.05 mol/L phosphate buffer (pH = 7) + 0.3 mol/L NaCl solution, and the flow rate was 1 mL/min. The detection wavelength was 200 nm. Bovine serum protein (67,000 Da), Vitamin B12 (1335 Da), and oxidized glutathione (614 Da) were used as the standard solution with a concentration of 5 mg/mL. The samples were filtered using a microporous membrane (0.45 μm) before sampling.

#### 2.3.3. Peptide Sequencing

An Orbitrap Fusion Lumos Mass Spectrometer was used to analyze the sequence of the peptides to obtain the original data, and then the peptide data was analyzed by database, or de novo, sequencing.

### 2.4. Determination of Antioxidant Capacity of RFP In Vitro

The same antioxidant can have different inhibitory effects on different free radicals. Accordingly, the antioxidant capacity of the samples was comprehensively evaluated by the clearance rate of the 1,1−Diphenyl−2−Picrylhydrazyl radical (DPPH^•^) [11], 2,2′−azino−BIS (3−ethylbenzothiazoline−6−Sulfonic acid) (ABTS) free radical (ABTS^•+^), superoxide anion (O^2•−^) [12], and hydroxyl radical (^•^OH) [13] in vitro. The Ferric reducing antioxidant power (FRAP) method was also employed to indirectly evaluate the antioxidant activity of samples. For specific operations, refer to the kit instructions.

### 2.5. Protective Effects of RFP on UVA Damaged Cells

#### 2.5.1. Cell Resuscitation and Culture

DMEM with 10% FBS and 1% Penicillin−Streptomycin Solution was prepared in advance. HSFs were removed from liquid nitrogen and immediately melted in a water bath at 37 °C. After being blown evenly, HSFs were immediately transferred to a culture flask containing an appropriate amount of medium and placed in a 5% CO_2_ incubator (Thermo Fisher Scientific Co., Ltd., Waltham, MA, USA) at 37 °C for resuscitation. After successful resuscitation, HSFs were washed gently in phosphate−buffered saline (PBS), and an appropriate amount of Trypsin−EDTA was added for digestion and passage, which was carried out when the cell density reached about 80%.

#### 2.5.2. Cell Viability Assay

The HSFs were inoculated in 96−well plates at a density of 8 × 10^3^ cells/well for 12 h, then covered with an appropriate amount of PBS and irradiated in an ultraviolet crosslinker (SCIENTZ03−II, Ningbo Scientz Biotechnology Co., Ltd., Ningbo, China). The UVA radiation doses were 0, 3, 6, 9, and 12 J/cm^2^. After irradiation, the HSFs were cultured in DMEM for 24 h, then incubated in 10 uL CCK−8 solution for 2–4 h. Absorbance was measured at 450 nm (Tecan M200 Infinite Pro, Tecan (Shanghai) Trading Co., Ltd., Shanghai, China) to determine the IC50 of the UVA−stimulated cell dose. RP and RFP were dissolved in DMEM at 20.00, 10.00, 5.00, 2.50, 1.25, 0.63, and 0.31 g/L. The HSFs were collected and cultured in 96−well plates for 12 h, then treated with sample solution replacement medium for 24 h, followed by incubation in DMEM containing 10% CCK−8 solution for 2–4 h. Absorbance was measured at 450 nm to evaluate the effects of the samples on cell viability. After the samples were treated for 24 h, the cells were irradiated with UVA at a dose of 12 J/cm^2^, cultured in DMEM for another 24 h, then incubated in 10 uL CCK−8 solution for 2–4 h. The absorbance was measured to evaluate the protective effects of the samples on UVA−damaged HSFs.

#### 2.5.3. Determination of ROS

The HSFs were inoculated in 6−well plates at a density of 1 × 10^6^ cells/well for 12 h. After sample treatment and UVA irradiation, they were treated with 2,7−dichlorodihydrofluorescein diacetate (DCFH−DA), which was hydrolyzed into 2,7−dichlorodihydrofluorescein (DCFH), then oxidized by ROS to 2,7−dichlorofluorescein (DCF) with fluorescence in intracellular incubation for 1 h. They were then washed in PBS, and the fluorescence intensity was measured (excitation wavelength of 485 nm and emission wavelength of 535 nm).

#### 2.5.4. Determination of MDA, Total Antioxidant Capacity, and Three Antioxidant Enzymes

The HSFs were laid in 6−well plates. After sample treatment and UVA irradiation, a 100 uL cell lysate buffer containing 1 mmol/L phenylmethanesulfonylfluoride (PMSF) was added to each well. After full lysis and 12,000× *g* centrifugation (Allegra X−30R, Beckman Coulter, Inc., Brea, CA, USA) at 4 °C for 5 min, the supernatant was taken, and the total antioxidant capacity, MDA content, and the activity of SOD, CAT, and GSH−Px were determined according to the kit instructions.

#### 2.5.5. Transcriptome Sequencing Analysis

After sample treatment and UVA irradiation, the total RNA of the cells was extracted using a trizol reagent. Next, 1% gel electrophoresis, ND−2000 (NanoDrop Technologies, Inc., Wilmington, DE, USA), and 2100 Bioanalyser (Agilent Technologies, Inc., Santa Clara, CA, USA) were used to analyze the degradation degree of 28S RNA and 18S RNA, determine whether there was any contamination, and detect the purity, concentration, and integrity of the total RNA to ensure its quality (OD260/280 = 1.8–2.2, OD260/230 ≥ 2.0, RIN ≥ 6.5, 28S: 18S ≥ 1.0, total >10 μg). The cDNA constructed according to the kit instructions was then enriched and quantified. Then, the Shanghai Majorbio Bio−Pharm Technology Co. Ltd. was commissioned to perform high−throughput sequencing on an Illumina HiSeq 4000 sequencing platform. With the adjusted *p*−value (P−adjust) < 0.05, |log2fold change| (|log2FC|) > 1.2 as the screening criteria, differential expression analysis was performed on the sequencing results using DESeq2 software. Goatools software was used for gene ontology (GO) enrichment analysis, and R script was used for the Kyoto Encyclopedia of Genes and Genomes (KEGG) pathway enrichment analysis. When P−adjust < 0.05, the GO function or KEGG Pathway was considered to be significantly enriched.

#### 2.5.6. RT−qPCR

According to the gene sequences published in NCBI, primers were designed by PrimerExpress software, and β−actin was used as the internal reference gene for RT−PCR validation (ABI7300, Thermo Fisher Scientific Co., Ltd.). The PCR conditions were as follows: 94 °C, 5 s; 60 °C, 15 s; 72 °C, 10 s; 94 °C, 5 s; 72 °C, 10 s; and the duration was 40 cycles. The relative expression levels were calculated using 2^−ΔΔCt^. The primer sequences are listed in Table 1.

#### 2.5.7. ELISA

The HSFs were laid in corning 25 cm^2^ rectangular neck cell culture flask with vent caps at a density of 5 × 10^6^ cells/flask for 12 h. After sample treatment and UVA irradiation, nuclear and cytoplasmic proteins were prepared according to the nuclear and cytoplasmic protein extraction kit. They were then used to determine the NRF2 Protein content using the Human Nuclear Factor E2−related Factor 2 (NRF2) ELISA Kit.

The HSFs were laid in corning 75 cm^2^ rectangular neck cell culture flask with vent caps at a density of 1.5 × 10^7^ cells/ flask for 12 h. After sample treatment and UVA irradiation, the preparation of cell lysates and the detection of the protein contents of NQO1 and HO−1 were performed according to the Human Quinone NADH Dehydrogenase 1 (NQO1) ELISA Kit and the Human Heme Oxygenase−1 (HO−1) ELISA Kit.

For specific test protocols, refer to the kit instructions.

### 2.6. Experiments on Animals

#### 2.6.1. Establishment of UVA Oxidative Damage Mouse Skin Model

Twenty female Kunming mice, weighing 18 ± 2 g, were purchased from Beijing Vital River Laboratory Animal Technology Co. Ltd. (Beijing, China) and randomly divided into the control group (double steam water), model group (double steam water + UVA), sample group (1 g/L RFP + UVA), and positive control group (1 g/L GSH+ UVA). Each set of five mice was housed in a separate cage, with the temperature kept between 22 °C and 25 °C. The circadian cycle consisted of 12 h of light and 12 h of darkness. They consumed food and liquids ad libitum. After one week of acclimation to the environment, the experiment was carried out. Before applying the samples, the back hair of the mice was shaved (3 × 3 cm) and they were allowed to adapt to the environment for 1 h. The sample volume was 1 mL/d, UVA irradiated the skin for 50 J/cm^2^/d, and the experiment lasted for 30 days [14].

#### 2.6.2. Tissue Sample Collection and Preparation

Tribromoethanol saturated solution was used for abdominal anesthesia, and 0.25–0.35 mL of venous blood was obtained by the tail−clipping method in an anticoagulant tube for routine blood level measurement. Afterwards, skin tissues removed from the same part of the mice’s back were divided into small pieces (100–150 mg), 1 mL of cell lysis solution was added, and the skin tissue was thoroughly homogenized using a glass homogenizer. The supernatant’s CAT activity and overall antioxidant capacity were assessed after centrifugation for 5 min (12,000× *g*). Other skin tissues were submerged in a 4% formaldehyde solution, gradient dehydrated with ethanol, and embedded in paraffin. The wax blocks were sliced into 4 m−thick slices, dewaxed, and traditionally rehydrated. They were then differentiated in 1% alcohol hydrochloride for 5 s after being soaked in hematoxylin for 5 min and washed in tap water for 1 h. Next, they were washed in tap water for 30 min following a 2 min period of saturation in a lithium carbonate aqueous solution. Lastly, they were sealed with neutral gum after being submerged in a 0.5% eosin solution for 2 min. Under a microscope, the structural traits of the skin tissues in each group were examined.

### 2.7. Statistical Analysis

Each experiment involved at least three biological repeats with three technical replicates. All variables were reported as mean ± standard deviation, and statistical significance between groups was determined by a univariate analysis of variance (ANOVA) test.

## 3. Results

### 3.1. Purification and Identification of Antioxidant Peptides from Rice Fermentation Broth

The first elution peak appeared at 130 mL, the second at about 470 mL, and the third at 710 mL. The content of the second peak component was low. The third peak had many impurities because the high concentration of salt solution is the regeneration condition of the column (Figure 1a). Consequently, the first component was selected for subsequent experiments. The gathered components were lyophilized after dialysis and named “RFP.” Compared with the HPLC chromatograms of the three standard substances (Figure 1b) and calculated according to the regression equation of the standard curve (with elution peak time as the abscissa, and the logarithm of relative molecular weight as the ordinate), the six peak areas of RFP were 22.25%, 20.30%, 29.18%, 8.72%, 4.82%, and 14.72%, and their molecular weights were 702.27 Da, 505.06 Da, 358.60 Da, 172.51 Da, 82.51 Da, and 44.76 Da, respectively (Figure 1c).

The results of peptide sequencing are presented in Table 2. RFP is a mixture composed of eight small peptides, each of which has fewer than 11 amino acids. The mean molecular weight of the amino acids was 128. The most abundant small peptide amino acid sequence is PLL, and the least abundant is YNEGDAPVVA. The amino acid composition of these small peptides is mostly hydrophobic amino acids (alanine (Ala, A), phenylalanine(Phe, F), isoleucine (Ile, I), leucine (Leu, L), proline (Pro, P), and valine (Val, V)). The amino acid sequences of YNDGDAPIVA, YNDGDAPIV, FYNDGDAPIV, and YNEGDAPVVA are relatively similar; except for hydrophobic amino acids, they all contain tyrosine (Try, Y), glycine (Gly, G), asparagine (Asn, N) and aspartate (Asp, D). All of the tyrosine and acidic amino acids (D or glutamate (Glu, E)) are separated by an N. RP consists of eight amino acids and the sequence is Lys−His−Asn−Arg−Gly−Asp−Glu−Phe. RP, YNDGDAPIVA, YNDGDAPIV, FYNDGDAPIV, and YNEGDAPVVA all contain N and G. RP and FYNDGDAPIV both contain F. Both RP and VRVF contain arginine (Arg, R). However, RP is mainly composed of basic amino acids (histidine (His, H), lysine (Lys, K) and R), and acidic amino acids (D, E).

### 3.2. In Vitro Antioxidant Capacity

In the concentration range of 2–10 g/L, the scavenging rates of RP with known antioxidant activity [15] and of RFP on DPPH^•^, ^•^OH, and O^2•−^ increased and stabilized with the increase in concentration (Figure 2a–c). The ABTS^•+^ scavenging rate and reduction ability were also good when the mass concentration of RFP and RP was 5 g/L (Figure 2d,e). Moreover, RFP had a more substantial scavenging rate of the above free radicals and reduction ability of iron ions than RP (*p* < 0.033). The half−maximal inhibitory concentration (IC50) of RFP was 4.269 g/L for DPPH^•^, and 3.806 g/L for ^•^OH.

### 3.3. Protective Effects of Antioxidant Peptides on UVA−Induced Oxidative Stress of HSF Cells

#### 3.3.1. Cell Viability

In the range of 0.31–20 g/L, RFP had no toxicity to HSFs and promoted cell proliferation, which was proportional to the concentration (Figure 3a). There was no toxicity to the cells when the concentration of RP was 1.25 g/L, at which level the cell survival rate was 82.01%. When the RP concentration was higher than 1.25 g/L, the cell survival rate dropped below 80%, showing certain cytotoxicity. Therefore, the concentration of RP and RFP in subsequent experiments was set at 1.25 g/L. The survival rate of HSFs decreased with the increase in UVA dose (Figure 3b). Low doses of UVA had no obvious inhibition effects on cell proliferation, while high doses of UVA caused great destruction to cells. When the UVA irradiation dose was 12 J/cm^2^, the survival rate decreased to 47.55 ± 0.81%, and the cell morphology changed from slender to round and even detached and dead. When pretreated with RP, RFP, and positive control (glutathione, GSH) for 24 h, followed by a 12 J/cm^2^ dose of UVA radiation, the survival rate and state of cells were significantly improved compared with the model group (Figure 3c). In addition, the cell survival rate of the RFP group was higher than that of the RP and positive control groups, with better cell morphology. Thus, RFP may reduce cell oxidative damage through cell proliferation.

#### 3.3.2. Antioxidation

RP, RFP, and GSH pretreatment significantly (*p* < 0.001) lowered the levels of ROS (Figure 4a) and MDA (Figure 4b) increased by UVA irradiation to normal levels, suggesting that all three prevent further oxidative stress damage by reducing the content of intracellular oxidative products. The total antioxidant capacity was further determined by ABTS and FRAP assays. It was found that the antioxidant capacity of cells irradiated with 12 J/cm^2^ UVA decreased significantly (*p* < 0.002), while RP, RFP, and GSH significantly (*p* > 0.033) enhanced the scavenging capacity of ABTS^•+^ (Figure 4c); In particular, the reduction capacity exceeded the normal level (Figure 4d), and the total antioxidant capacity of RFP was finer than that of GSH and RP (*p* < 0.002).

#### 3.3.3. Analysis of Protective Mechanism of RFP against UVA Oxidative Stress in HSF Cells Based on Transcriptome

A total of 1,790,640,814 clean reads were obtained by constructing a transcriptional library from nine samples (Table A1). The average percentage of Q20 bases was 98.69%, that of Q3 bases was 95.59%, G and C bases accounted for 47.81% of the total number of bases, and the mapped total was 65.70%, indicating that the quality of the sequencing results is positive. The correlations between the three types of samples were analyzed and the results showed that the correlations between the three biological replicates of each sample were strong (Figure 5a). This implies that the data obtained by the transcriptome was exact and reliable, and that it can be investigated in the next step. Compared with the control group, 1875 genes were significantly differentially expressed in the model group, including 725 up−regulated genes and 1147 down−regulated genes (Figure 5b). Compared with the model group, 9792 genes were significantly differentially expressed in the RFP group, including 5007 up−regulated genes and 4785 down−regulated genes (Figure 5c). GO biological process analysis was performed on the differential genes in the three groups of cell samples, and a total of 1927 items were screened under *p* < 0.05 conditions. The initial 20 were significantly enriched and highly correlated with stress, such as the regulation of DNA−templated transcription in response to stress (Figure 5d). A total of 331 pathways were enriched by KEGG enrichment analysis conducted on particular signaling pathways. Among the top 30 pathways with significant enrichment, the number of genes in cancer pathways was the largest (Figure 5e). In addition, significantly up−regulated and down−regulated genes related to the NRF2 oxidative stress signaling pathway were found when processing the endoplasmic reticulum, hepatocellular carcinoma, and cancer pathways (Table A2).

#### 3.3.4. Quantitative Expression Analysis

The mRNA expression of KEAP1 was significantly (*p* < 0.001) increased and NRF2 was significantly (*p* < 0.001) decreased after UVA irradiation, and the UVA effects were inhibited in both the high− and low−dose RFP groups (Figure 6a,b). In addition, the protein content of NRF2 in the cytoplasm was significantly (*p* < 0.002) decreased, and the protein content in the nucleus was increased to a certain extent by RFP pre−treatment (Figure 6c). However, the effects of RFP were inhibited in the presence of NRF2 inhibitor all−trans−retinoic acid (ATRA) [16].

The relative expression of glutathione S−transferase mu 1 (GSTM1) was significantly (*p* < 0.001) increased in both the high− (4 g/L) and low−dose (1 g/L) RFP groups (Figure 7a), and the expression of glutathione S−transferase theta 1 (GSTT1) in the high−dose group was significantly (*p* < 0.033) increased (Figure 7b). The expression levels of glutamate−cysteine ligase modifier subunit (GCLM) and glutamate−cysteine ligase catalytic subunit (GCLC) rose with the increase in RFP concentration (Figure 7c,d). In the presence of ATRA, the effects of RFP were inactivated, resulting in the inhibition of the relative expression levels of GSTM1, GSTT1, GCLM, and GCLC, which were comparable to or even lower than those in the UVA−irradiated group.

The expression levels of HO−1 (Figure 8a), SOD (Figure 8e), and GSH−Px (Figure 8i) were markedly (*p* < 0.001) increased in the RFP high−dose group. The expressions of NQO1 (Figure 8c) and CAT (Figure 8g) in high− and low−dose groups were enhanced, and the effects on the high−dose group were superior. In the presence of ATRA, the pretreatment effects of RFP were also lost, and the relative expression level of each enzyme was inhibited. Further quantitative analysis of the enzyme activity of the five enzymes indicates that RFP promotes the enzyme activity that is reduced by UVA radiation (Figure 8b,d,f,h,j).

### 3.4. Protective Mechanism of RFP against Oxidative Stress in Mice

#### 3.4.1. Blood Routine Test Results

It can be seen from Table A3 that when treating and UVA irradiating mice for two months, there were no significant differences in the white blood cells, red blood cells, platelets, hemoglobin, neutrophils, and lymphocytes among the groups, and the measured values were within the normal range, showing that GSH and RFP had no side effects on the immune system of mice.

#### 3.4.2. Skin Histological Observation

After 30 days of UVA radiation, the skin of mice in the model group wrinkled and became less elastic compared to the blank group. Tissue sections stained by H&E likewise demonstrated that the epidermis was smooth and the dermal collagen fibers were tightly arranged in the control group, showing that the experimental hair removal did not alter the effect on the fundamental makeup of the skin tissue (Figure 9a). In the model group, there were more deep folds in the back epidermis, irregular proliferation, hyperkeratosis and more shedding in the cuticle. The collagen fiber bundles (pink) in the dermis were diminished, distorted, torn and disordered (Figure 9b). Compared with the model group, the epidermis of the GSH (Figure 9c) and RFP (Figure 9d) groups was smoother, the cuticle hyperplasia was not obvious, and the collagen in the dermis was more tightly arranged. These results indicate that RFP improves the appearance of skin photoaging caused by UVA radiation to a certain extent.

#### 3.4.3. Total Antioxidant Capacity and CAT Enzyme Activity in Mice Tissue

After being irradiated with UVA for a long time, the antioxidant capacity and CAT enzyme activity of the skin of mice in the model group decreased to an extent similar to the results of the cell antioxidant test, while RFP and positive control GSH markedly (*p* < 0.033) increased CAT enzyme activity and antioxidant capacity (Figure 9e,f).

## 4. Discussion

With the continuous deterioration of the ecological environment, human skin exposed to UV radiation for an extended period of time or in high doses can produce local inflammation, oxidative stress, abnormal pigmentation, and even skin cancer [17]. Compared with UVB, UVA is more likely to induce oxidative stress damage and is more cytotoxic and carcinogenic [18]. The in vitro skin model exposed to excessive UVA radiation generates high levels of ROS that are dose−dependent on UVA and result in oxidative stress [19]. In this work, it was discovered that excessive ROS generation caused an imbalance in the antioxidant system, using HSF cells and mouse skin oxidative stress models induced by UVA. Antioxidant supplementation successfully halts the advancement of several illnesses brought on by oxidative stress [20]. Indirect antioxidants are those that take part in the activation of endogenous antioxidant signaling pathways, thereby promoting the transcriptional expression of a wide range of cellular protective genes that resist oxidative stress, while direct antioxidants directly react with free radicals through their structure to break the chain of free−radical reactions [21]. Natural products have been widely studied for their advantages, which include low toxicity and high biological activity. The polypeptide fraction (535–2959 Da) obtained by the alkaline protease hydrolysis of cocoa bean gluten shows high scavenging activity of DPPH^•^ and ABTS^•+^ [22]. The antioxidant peptide obtained from Tegillarca granosa scavenges DPPH^•^, ^•^OH, and O^2•−^, but also scavenges the excessive ROS produced by oxidative stress in PC−3 cells [23]. The antioxidant peptide prepared by the Bacillus subtilis solid−state fermentation of corn gluten powder improves the activities of SOD, CAT, and GSH−Px in the serum and liver of D−galactose−induced aging mice [24]. DPPH^•^ and ABTS are synthetic macromolecules. ABTS oxidizes to ABTS^•+^ under the action of an appropriate oxidant and appears green. O^2•−^ has oxidative toxicity for cells and is strongly linked to the formation of other ROS in organisms [25]. ^•^OH is the most active free radical and has the strongest oxidative damage effect on biological macromolecules [26]. SOD, CAT, and GSH−Px are the three primary enzymes in the cellular enzymatic antioxidant system, cooperating with each other to form an antioxidant chain. To prevent the formation of the more destructive ^•^OH, SOD first converts O^2•−^ to H_2_O_2_, and then CAT and GSH−Px decompose H_2_O_2_ into water and oxygen. GSH−Px also converts lipid peroxides into nontoxic products [27]. These above results indicate that antioxidant peptides are good bifunctional antioxidants.

The antioxidant activity of antioxidant peptides is closely related to their molecular weight, amino acid composition, and sequence. As they are more likely to interact with target radicals to end their chain reactions, peptides with lower molecular weight generally have stronger antioxidant activity than peptides with higher molecular weight [28]. Most antioxidant peptides with molecular weights of 500–1800 Da and which contain 2–9 amino acids have stronger antioxidant capacities than macromolecular proteins [8]. The peptides in RFP obtained from *Lactobacillus plantarum* rice fermentation broth were all composed of 10 or fewer amino acids with 300–1200 Da (Table 2). The activity of antioxidant peptides containing hydrophobic amino acids or acid−base amino acids is enhanced. Peptides containing hydrophobic amino acids upgrade their solubility on the lipid−water surface through fatty side chains, and eliminate free radicals by better employing addition reactions [29]. Moreover, the hydrophobicity of N−terminal amino acid is positively correlated with antioxidant capacity, and the antioxidant activity of peptides is generally stronger when L or V are located at the N−terminal amino acid [7]. Acidic or basic amino acids also play an important role in the antioxidant activity of peptides that utilize the carboxyl or amino groups on the side chains as metal ion chelators [30]. Each peptide in RFP contains more than three hydrophobic amino acids. The N−terminal of LLLP, VRVF, and LLSPF3 peptides is L or V. In addition, five peptides contain acid and basic amino acids. Y aromatic groups and phenolic hydroxyl structures can provide protons to quench free radicals. Additionally, if Ds and Es that contain carboxylic acid roots are close to Y, the oxygen electron cloud density on the Y phenolic hydroxyl group is reduced, which is more conducive to proton release and enhances the proton donor effect of Y [28]. The N−terminus of YNDGDAPIVA, YNDGDAPIV, and YNEGDAPVVA is Y, and D and E are present adjacent. Therefore, it can be inferred that RFP has great antioxidant activity. In vitro antioxidant experiments demonstrated that RFP is a direct antioxidant superior to RP, with better ability to scavenge free radicals (DPPH^•^, O^2•−^, ^•^OH, and ABTS^•+^) and reduce iron ions (Figure 2a–e).

Subsequently, we conducted a UVA−induced oxidative stress experiment on HSF cells and found that RFP significantly reduces the content of ROS and MDA, which are the hallmark products of cellular oxidative stress damage [31,32], and increases the total antioxidant capacity (Figure 4).

Transcriptome analysis, which reflects genes that are actively expressed under specific conditions, was further performed. This is an important means of studying cell functions and phenotypes which has a lower cost than Sanger sequencing, good accuracy in terms of expression levels, and whose results display a high level of repeatability. Differentially expressed genes related to the NRF2 antioxidant pathway were observed in the sequencing results (Table A2). The KEAP1−NRF2/ARE signaling pathway is important for maintaining the redox balance in cells. Acting as a redox damage sensor, KEAP1 binds with effector transcription factor NRF2 in the N−terminal Neh2 region under the action of ubiquitin ligase 3 to form a dimer which remains in the cytoplasm for rapid decomposition, preventing NRF2 from entering the nucleus for transcription under normal physiological conditions. When buoyed by oxidative stress, NRF2 dissociates from KEAP1, transfers into the nucleus and binds to the specific sequence MAF on the DNA to form a heterodimer that recognizes and binds to the antioxidant response element in the promoter region of Phase II detoxification enzyme genes, thereby regulating the expression of downstream Phase II detoxification enzymes, maintaining the redox balance in the cells, and improving the ability of the cells to resist oxidative stress [33]. Besides the three main antioxidant enzymes, NRF2−related Phase II detoxification enzymes include glutathione glutamate−cysteine ligase (GCL), S−transferases (GST), NQO1, and HO−1. GCL is the rate−limiting enzyme of GSH synthesis in vivo, consisting of GCLC and GCLM, which are up−regulated to promote the synthesis of GSH and perform an important regulatory role in the body’s antioxidant defense [34]. GSTT1 and GSTM1 are members of GST whose gene expression products promote electrophilic substances and GSH to form more hydrophilic products for better metabolism to defend macromolecules from attack [35]. HO−1, an important rate−limiting enzyme in heme metabolism, has a negative regulatory effect on oxidative stress, and its increased expression improves its antioxidant capacity. NQO1 reduces quinones to reduce the oxygen radicals produced by quinone transformation. In a study of myocardial infarction induced in mice by isoproterenol and H9c2 cell hypoxia and reoxygenation injury, it was found that hirudin plays a cell protection role, activates the NRF2 signaling pathway by interfering with the KEAP1−NRF2 complex so that NRF2 is transferred from the cytoplasm to the nucleus, and then regulates NRF2−dependent protein gene expression [36]. In the process of the RT−PCR and protein quantitative verification of the transcriptome results, we detected that RFP similarly promotes NRF2 transcription and protein expression, which is identical to the transcriptome results; thus, RFP promotes nuclear translocation, and its role is limited in the presence of NRF2 inhibitors (Figure 6a–c). In addition, RFP significantly up−regulates the transcription levels of GSTM1, GSTT1, GCLC, and GCLM, which have been down−regulated by UVA irradiation in cells through the NRF2 pathway (Figure 7a–d). The transcription and protein levels of HO−1, NQO1, SOD, GSH−Px, and CAT were similarly elevated (Figure 8). In the UVA−induced mouse skin oxidative stress model, we also discovered that RFP enhances the total antioxidant capacity and CAT enzyme activity of mouse skin tissues (Figure 9). RFP is decidedly an effective bifunctional antioxidant that plays an important role in opposing oxidative stress.

## 5. Conclusions

RFP isolated and purified from *Lactobacillus plantarum* rice fermentation broth is a blend of small peptides with 10 amino acids or less, which shows positive antioxidant activity at the chemical model, cell biology, and animal model levels. However, different peptides have different antioxidant mechanisms. Therefore, the further isolation and purification of RFP should be carried out in subsequent experiments to verify the antioxidant effectiveness of single peptides one by one.

## Figures and Tables

**Figure 1 antioxidants-11-02333-f001:**
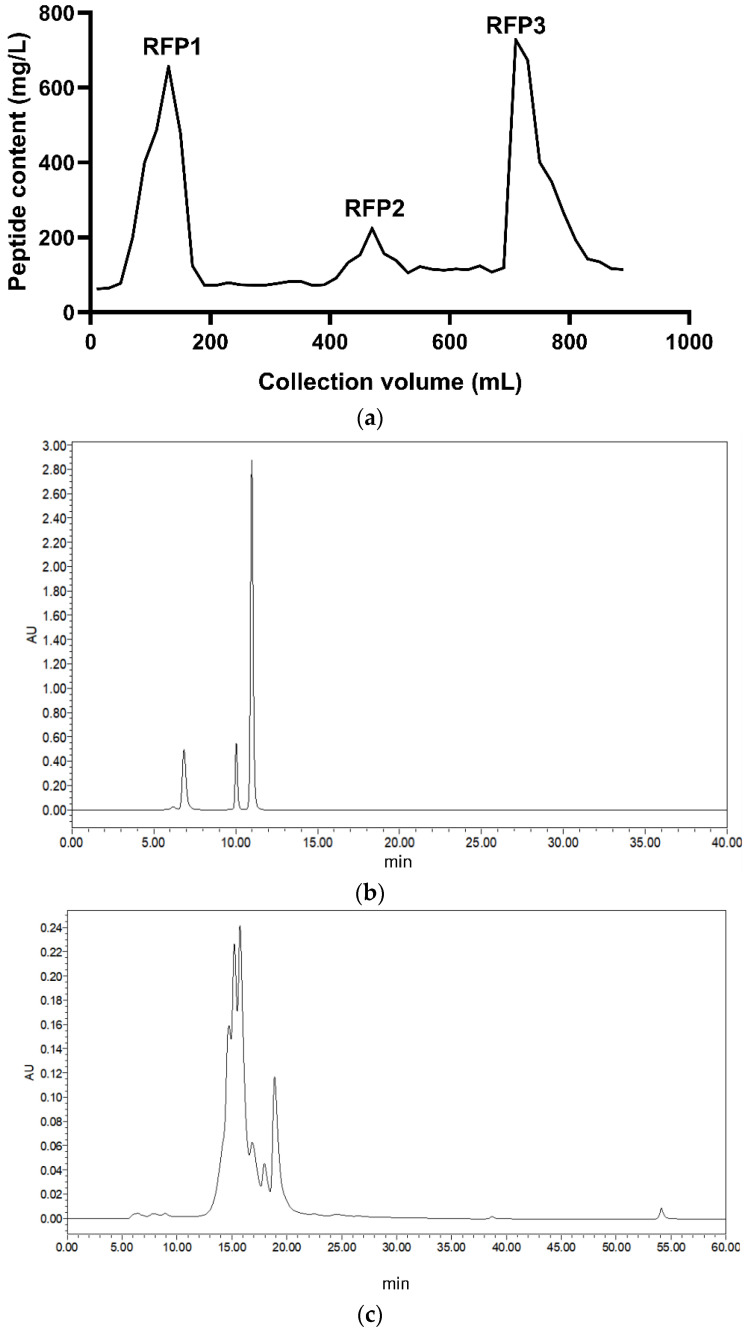
Purification of peptides from fermentation broth: (**a**) Purification curves of rice fermentation peptides in chromatography column; (**b**) elution curves of RP−HPLC for standard samples; and (**c**) elution curves of RP−HPLC for RFP.

**Figure 2 antioxidants-11-02333-f002:**
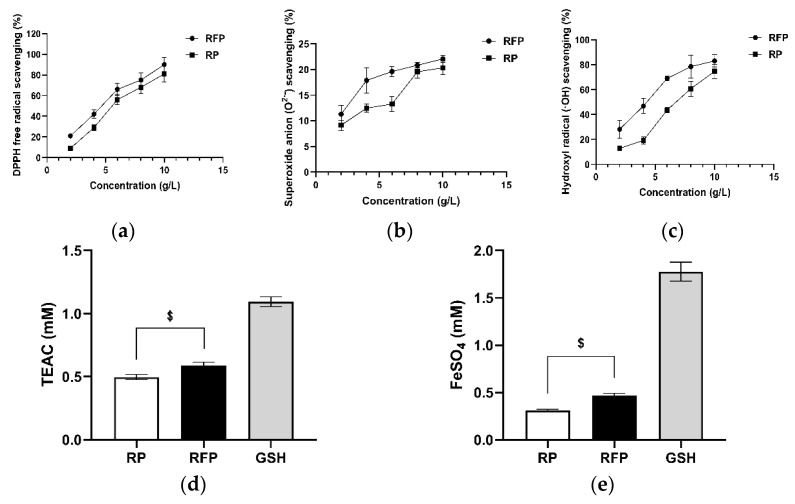
Antioxidant capacity of RFP determined by chemical model method: (**a**) scavenging rate of DPPH^•^; (**b**) scavenging rate of O^2•−^; (**c**) scavenging rate of ^•^OH; (**d**) total antioxidant capacity (ABTS assay); and (**e**) total antioxidant capacity (FRAP assay). $ *p* < 0.033 as compared to RP group. Values do not have a common mark ($) when *p* > 0.033, the same below.

**Figure 3 antioxidants-11-02333-f003:**
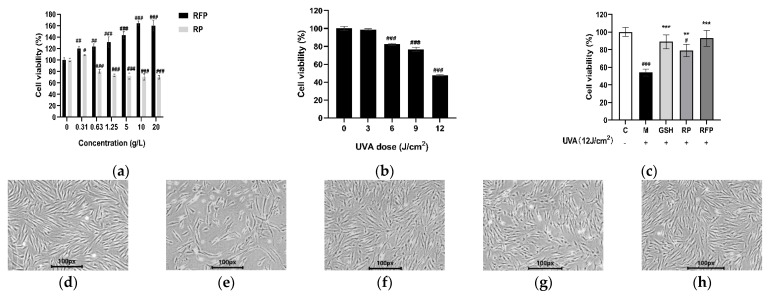
Effects of UVA, RFP, and RP on survival rate and morphology of HSF: (**a**) effects of RP and RFP on proliferation of HSF cells; (**b**) effects of different doses of UVA on survival rate of HSF cells; (**c**) cell survival rate of HSFs protected by samples; (**d**) control group; (**e**) model group; (**f**) positive group; (**g**) RP group; and (**h**) RFP group. # *p* < 0.033, ## *p* < 0.002, ### *p* < 0.001 as compared to control group (C); ** *p* < 0.002, *** *p* < 0.001 as compared to damage model group (M). Values do not have a common mark (#, *) when *p* > 0.033.

**Figure 4 antioxidants-11-02333-f004:**
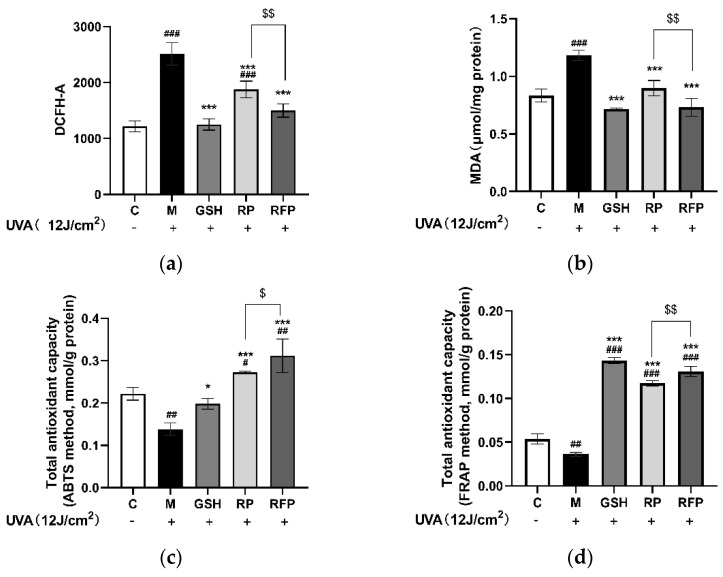
Antioxidant capacity of RFP determined by cell biology model: (**a**) effects on ROS content; (**b**) effects on MDA content; (**c**) effects on total antioxidant capacity (ABTS assay); and (**d**) effects on total antioxidant capacity (FRAP assay). # *p* < 0.033, ## *p* < 0.002, ### *p* < 0.001 as compared to control group (C); **p* < 0.033, *** *p* < 0.001 as compared to damage model group (M); $ *p* < 0.033, $$ *p* < 0.002 as compared to RP group; Values do not have a common mark (#, *, $) when *p* > 0.033.

**Figure 5 antioxidants-11-02333-f005:**
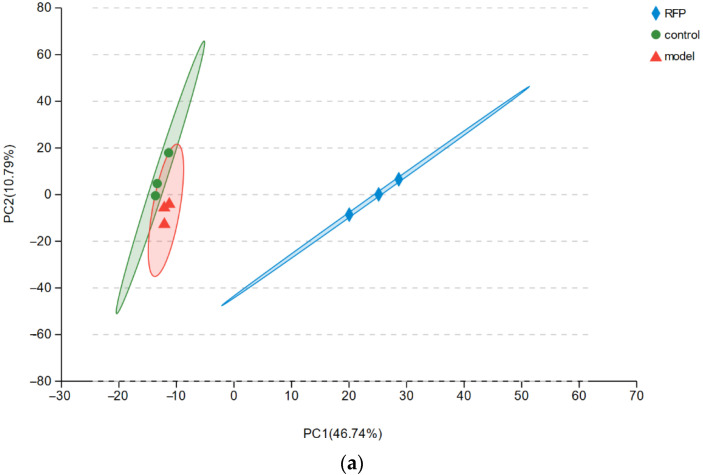
Transcriptome sequencing analysis: (**a**) principal component analysis (PCA); (**b**) differentially expressed genes between control and model groups; (**c**) differentially expressed genes between model and RFP groups; (**d**) results of GO functional enrichment of the differentially expressed genes of three groups; and (**e**) results of KEGG functional enrichment of the differentially expressed genes of three groups.

**Figure 6 antioxidants-11-02333-f006:**
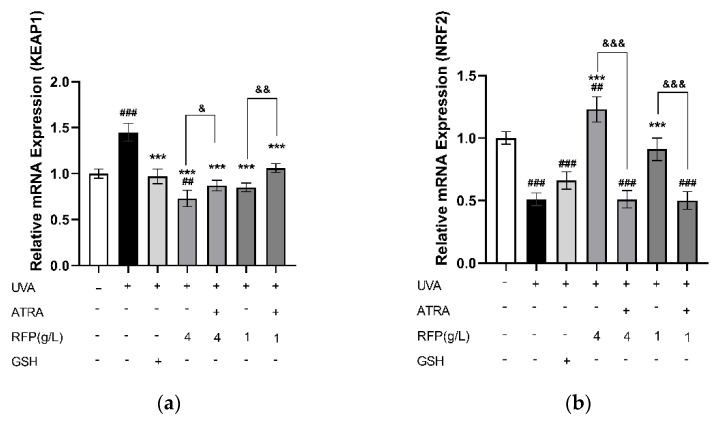
Effects of RFP on nuclear translocation of NRF2: (**a**) effect on relative expression of KEAP1; (**b**) effect on relative expression of NRF2; and (**c**) effect on protein content of cytoplasm and nucleus NRF2. # *p* < 0.033, ## *p* < 0.002, ### *p* < 0.001 as compared to control group; ** *p* < 0.002, *** *p* < 0.001 as compared to damage model group; & *p* < 0.033, && *p* < 0.002, &&& *p* < 0.001 as compared to the group with ATRA. Values do not have a common mark (#, *, &) when *p* > 0.033.

**Figure 7 antioxidants-11-02333-f007:**
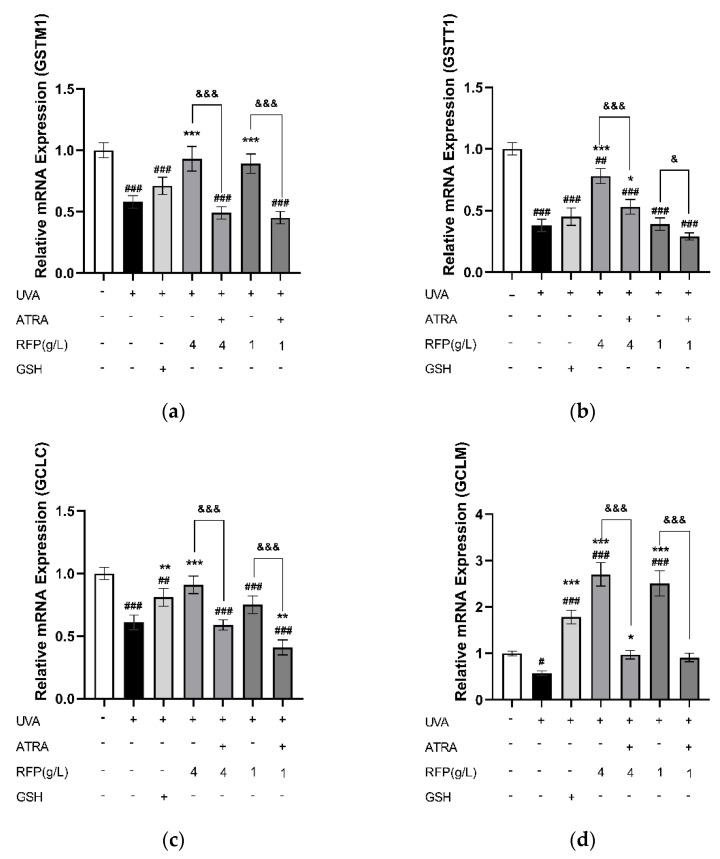
Effects of RFP on relative expression of GST and GCL genes: (**a**) effect on GSTM1; (**b**) effect on GSTT1; (**c**) effect on GCLC; and (**d**) effect on GCLM. # *p* < 0.033, ## *p* < 0.002, ### *p* < 0.001 as compared to control group; * *p* < 0.033, ** *p* < 0.002, *** *p* < 0.001 as compared to damage model group; & *p* < 0.033, &&& *p* < 0.001 as compared to the group with ATRA. Values do not have a common mark (#, *, &) when *p* > 0.033.

**Figure 8 antioxidants-11-02333-f008:**
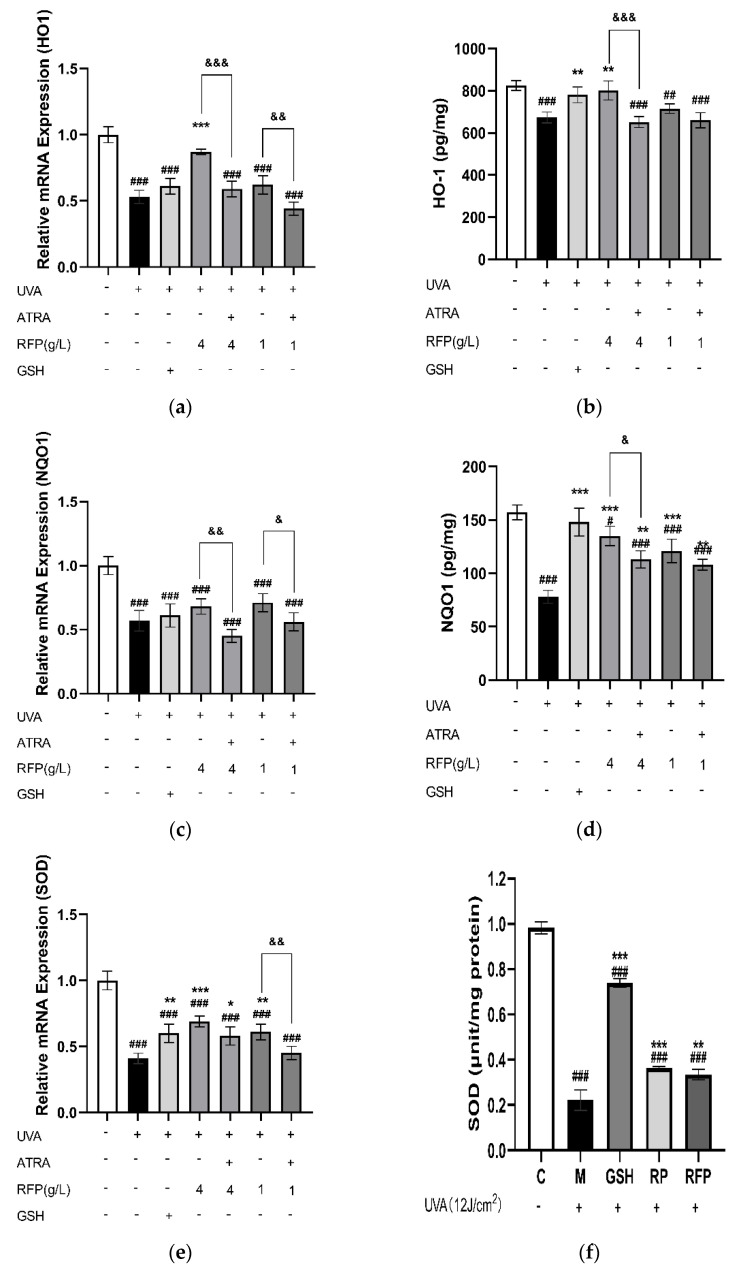
Effects of RFP on transcription expression and content of antioxidant enzymes: (**a**) effect on transcription expression of HO−1; (**b**) effect on content of HO−1; (**c**) effect on transcription expression of NQO1; (**d**) effect on NQO1content; (**e**) effect on transcription expression of SOD; (**f**) effect on SOD content; (**g**) effect on transcription expression of CAT; (**h**) effect on CAT content; (**i**) effect on transcription expression of GSH−Px; and (**j**) effect on GSH−Px content. # *p* < 0.033, ## *p* < 0.002, ### *p* < 0.001 as compared to control group; * *p* < 0.033, ** *p* < 0.002, *** *p* < 0.001 as compared to damage model group; & *p* < 0.033, && *p* < 0.002, &&& *p* < 0.001 as compared to the group with ATRA. Values do not have a common mark (#, *, &) when *p* > 0.033.

**Figure 9 antioxidants-11-02333-f009:**
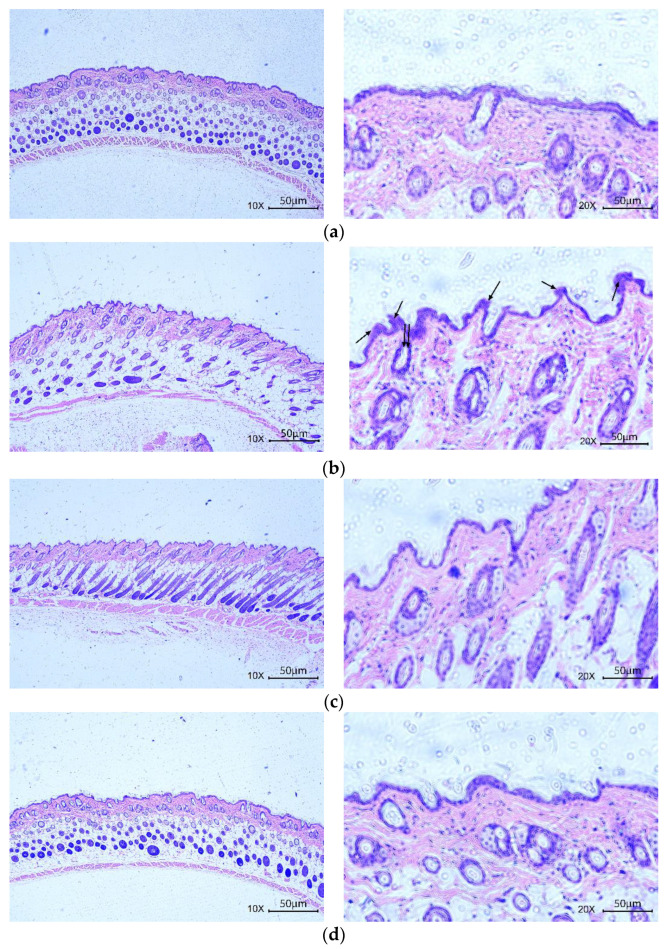
Effects of RFP on H&E staining and antioxidant capacity of mice skin tissues: (**a**) control group (double steam water); (**b**) model group (double steam water + UVA); (**c**) positive control group (1 g/L GSH + UVA); (**d**) sample group (1 g/L RFP + UVA); (**e**) effect on antioxidant capacity; and (**f**) effect on CAT activity. Single arrow indicates irregular epidermal hyperplasia; the double arrow points to the corner plug of the hair follicle. ## *p* < 0.002, ### *p* < 0.001 as compared to control group; * *p* < 0.033, *** *p* < 0.001 as compared to damage model group. Values do not have a common mark (#, *) when *p* > 0.033.

**Table 1 antioxidants-11-02333-t001:** Primer sequences for real−time PCR.

Gene	Direction	Primer Pair Sequence (5′→3′)
*β−actin*	F	TGGCACCCAGCACAATGAA
	R	CTAAGTCATAGTCCGCCTAGAAGCA
*NRF2*	F	CAACTCAGCACCTTGTATC
	R	TTCTTAGTATCTGGCTTCTT
*KEAP1*	F	GGAGGCGGAGCCCGA
	R	GATGCCCTCAATGGACACCA
*HO−1*	F	CAAGCGCTATGTTCAGCGAC
	R	GCTTGAACTTGGTGGCACTG
*NQO1*	F	CAGCCAATCAGCGTTCGGTA
	R	CTTCATGGCGTAGTTGAATGATGTC
*GCLC*	F	CAGTCAAGGACCGGCACAAG
	R	CAAGAACATCGCCTCCATTCAG
*GCLM*	F	TAAATCCCGATGAAAGAG
	R	AACAGGAGGTGAAGCAAT
*GSTM1*	F	GAACTCCCTGAAAAGCTAAAGC
	R	GTTGGGCTCAAATATACGGTGG
*GSTT1*	F	TTCCTTACTGGTCCTCACATCTC
	R	TCACCGGATCATGGCCAGGCGCA
*SOD*	F	TGGAGATAATACAGCAGGCT
	R	AGTCACATTGCCCAAGTCTC
*CAT*	F	CCTTCGACCCAAGCAA
	R	CGATGGCGGTGAGTGT
*GSH−Px*	F	AGAAGTGCGAGGTGAACGGT
	R	CCCACCAGGAACTTCTCAAA

**Table 2 antioxidants-11-02333-t002:** Peptide sequencing of RFP.

Sequence	Length	Mass (Da)
PLL	3	341.23146
YNDGDAPIVA	10	1033.4716
LLLP	4	454.31552
YNDGDAPIV	9	962.43453
VRVF	5	575.3319
LLSPF	4	519.31692
FYNDGDAPIV	10	1109.5029
YNEGDAPVVA	10	1033.4716

## Data Availability

The data presented in the study are available in the article.

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
