# Peer review of "Purification and Identification of Antioxidant Peptides from Rice Fermentation of Lactobacillus plantarum and Their Protective Effects on UVA−Induced Oxidative Stress in Skin"

_antioxidants, 2022, doi:10.3390/antiox11122333_

Round 1
Reviewer 1 Report
Title: Purification and Identification of Antioxidant Peptides from 2 Rice Fermentation of Lactobacillus plantarum and their Protective Effects on UVA-induced Oxidative Stress in Skin
A significant novelty and interest has been found in the research article, which will be helpful to those who are working in this field. In the submitted article, the objectives and aims are clearly defined and consistent with the journal's objectives. In order to confirm the antioxidant properties of isolated peptides, the authors have experimented with a variety of methods and generated several experimental results. Nevertheless, authors must concentrate on improving their manuscript in a number of ways in order to obtain good scientific publication.
Authors should focus on improving the manuscript in several ways at the scientific publication level.
Present all presentations in past tense instead of present tense
Thus, in this study, rice was fermented by Lactobacillus plantarum, and rice fermentation polypeptides (RFP) were extracted and purified from the fermentation broth.
These lines are not clear. RFP consists of peptides with fewer than 11 amino acids which enhances the expression of the antioxidant enzyme 22 genes downstream of the KEAP1-NRF2 / ARE pathway by promoting nuclear factor-erythroid related factor 2 (NRF2) nuclear translocation to remove lipid oxidation products and excess free radicals.
These lines are not clear. Nevertheless, the fermentation method employing protease pro-47 duced by microbial species in the process of growth and metabolism to hydrolyze sub-48 strates and prepare bioactive peptides is straightforward, has low production cost, and is 49 easily industrialized.
These lines are not clear. However, because of the intricate oxidation process, 58 whether antioxidant peptides with high antioxidant activity in vitro have the same impact 59 in vivo calls for more analysis together with biological tests.
Authors have to clearly represent the aim and objectives of the presented work at the end of the introduction section.
Material and Methods
All the experimental protocols and their execution are not well as scientific level.
Authors have request to improve the experimental part in all ways such as cultivation of L.Plantarum and its conditions, how much bacterial colonies were inoculated for fermentation process (CFU/mL), alcohol ratio used to precipitate polysachharides, freeze drying conditions and its final yield. Sample loading to column, how much sample/ minute how much water and NaCl/minute, cell harvesting ( trypsin alone and Trypsin-EDTA), Cell culture experiments, transcriptome sequencing details ( company name, tool and software used to detect the signaling pathways, etc.
Results and discussion
This is not clear sentences. The average molecular weight of the amino acids was 128. The peptide sequencing 239 results show that RFP is a mixture of small peptides with less than 11 peptides, and the 240 contents of each component decrease from top to bottom. RFP is a mixture of small peptides with less than 11 amino acids?
It is not acceptable to publish experimental results and their presentation as scientific publications. The authors requested that the quality of the presentation of the data as well as the interpretation of the results be improved.
The overall language of the manuscript is not well written.
The name of the microorganism must be in italics
Author Response
Oct. 20, 2022
Manuscript ID: antioxidants-1970346
Type of manuscript: Article
Title: Purification and Identification of Antioxidant Peptides from Rice
Fermentation of Lactobacillus plantarum and their Protective Effects on
UVA-induced Oxidative Stress in Skin
Authors: Qiuting Mo, Shiquan You, Hao Fu, Dongdong Wang, Jiachan Zhang,
Changtao Wang, Meng Li *
Received: 29 September 2022
E-mails: 2130041028@st.btbu.edu.cn, shiquan.you@starealife.com, 20111006@th.btbu.edu.cn, wdd@btbu.edu.cn, 20120720@btbu.edu.cn, wangct@th.btbu.edu.cn, limeng@btbu.edu.cn
Dear referee,
Thank you very much for giving us this opportunity to revise our manuscript. The comments and suggestions made on our manuscript are very encouraging and helpful. Detailed point-by-point responses to all comments are provided in the following pages. Note that the comments are presented in italics, and our responses are in Roman and blue font. All changes made to the manuscript were marked using the “Track Changes” function. In addition, we addressed all these major points and other issues carefully and revised the manuscript accordingly. Please let me know if you have any further questions.
Sincerely,
Meng Li
Beijing Key Lab of Plant Resource Research and Development, College of Chemistry and Materials Engineering, Beijing Technology and Business University, Fucheng Road, Beijing 100048, China
Tel.: +86-13426015179
E-mail: limeng@btbu.edu.cn
Title & Abstract
- Present all presentations in past tense instead of present tense.
Reply: Thanks for giving us the opportunity to revise the manuscript and your comments. We have changed the present tense into the past tense.
“Thus, in this study, rice was fermented by Lactobacillus plantarum, and rice fermentation polypeptides (RFP) consisting of peptides with fewer than 11 amino acids were extracted and purified from the fermentation broth. Subsequently, the antioxidant activity of RFP was assessed through the chemical model, cell biology, and animal model methods.”
- (1) These lines are not clear. RFP consists of peptides with fewer than 11 amino acids which enhances the expression of the antioxidant enzyme 22 genes downstream of the KEAP1-NRF2 / ARE pathway by promoting nuclear factor-erythroid related factor 2 (NRF2) nuclear translocation to remove lipid oxidation products and excess free radicals.
(2) These lines are not clear. Nevertheless, the fermentation method employing protease pro-47 duced by microbial species in the process of growth and metabolism to hydrolyze sub-48 strates and prepare bioactive peptides is straightforward, has low production cost, and is 49 easily industrialized.
(3) These lines are not clear. However, because of the intricate oxidation process, 58 whether antioxidant peptides with high antioxidant activity in vitro have the same impact 59 in vivo calls for more analysis together with biological tests.
Reply: Thanks for pointing out the questions. We have reformulated the main idea of these sentences.
“Thus, in this study, rice was fermented by Lactobacillus plantarum, and rice fermentation peptides (RFP) which is a mixture of small peptides with less than 11 amino acids were extracted and purified from the fermentation broth. Subsequently, the antioxidant activity of RFP was assessed through the chemical model, cell biology, and animal model methods. RFP enhanced the expression of the antioxidant enzyme genes downstream of the KEAP1-NRF2/ARE pathway by promoting nuclear factor-erythroid 2-related factor 2 (NRF2) nuclear translocation while simultaneously removing lipid oxidation products and excess free radicals. These results suggest that RFP is a potential substance for resisting aging and disease caused by oxidative stress.”
“Nevertheless, the preparation of bioactive peptides by microbial fermentation has the advantages of simplicity, low production cost and easy industrialization.”
“However, peptides with strong antioxidant activity at the biochemical level do not necessarily have the same effect on organisms. Therefore, it is necessary to combine biological tests for evaluation based on chemical method evaluation.”
- Authors have to clearly represent the aim and objectives of the presented work at the end of the introduction section.
Reply: Thanks for your comments and your great suggestions. We have added the purpose and objectives of the presented work at the end of the introduction.
“To examine the antioxidant activity of the rice fermentation polypeptides (RFP) extracted from rice fermentation broth fermented by Lactobacillus plantarum. Firstly, the chemical model determined the free radical scavenging and ferric reduction ability of RFP. Then, the effects of RFP on the scavenging ability of reactive oxygen species (ROS), malondialdehyde (MDA) and the overall antioxidant capacity of human skin fibroblasts (HSF) induced by UVA oxidative stress were verified. The effects of RFP on KEAP1-NRF2 / ARE pathway were also confirmed. Finally, we preliminarily detected the effects of RFP on catalase (CAT) enzyme and total antioxidant capacity in UVA-irradiated mice, and observed the changes of mouse skin through tissue sections.”
Material and Methods
Authors have request to improve the experimental part in all ways such as cultivation of L. Plantarum and its conditions, how much bacterial colonies were inoculated for fermentation process (CFU/mL), alcohol ratio used to precipitate polysachharides, freeze drying conditions and its final yield. Sample loading to column, how much sample/ minute how much water and NaCl/minute, cell harvesting (trypsin alone and Trypsin-EDTA), Cell culture experiments, transcriptome sequencing details (company name, tool and software used to detect the signaling pathways, etc.
Reply: Thanks for giving us the opportunity to revise the manuscript and pointing out the question. We have added or modified the problems you pointed out.
“The MRS medium was configured and sterilized by autoclaving for 15min. After cooling to room temperature, Lactobacillus plantarum suspension was added for expansion culture. The culture conditions were 37℃ and 180rpm for 24h.”
“Deionized water was added at a material-to-liquid ratio of 1:4. After sterilization, 5% activated Lactobacillus plantarum liquid (107 CFU/mL) was added, and fermentation was performed at 180 r/min at 37℃.”
“After centrifugation, the polysaccharides of the supernatant were removed by the alcohol precipitation method and the ratio of ethanol to supernatant was 4:1. After freeze-drying at -60℃ in vacuum rotation, the freeze-dried rice fermentation liquid powder was obtained and the yield was 0.17 g/L.”
“The sample loading volume was 2 mL and the flow rate was 2 mL/min. They were then eluted sequentially with deionized water, 0.1 NaCl solution, and 1 mol/L NaCl solution at a flow rate of 2 mL/min.”
“After successful resuscitation, HSF were washed gently in phosphate-buffered saline (PBS), and an appropriate amount of Trypsin-EDTA was added for digestion and passage, which was carried out when the cell density reached about 80%.”
“After being blown evenly, HSF were immediately transferred to a culture flask containing an appropriate amount of medium and placed in a 5% CO2 incubator (Thermo Fisher Scientific Co., Ltd.) at 37℃ for resuscitation.”
“HSF were inoculated in 96-well plates at a density of 8 × 103 cells/well for 12 h, then covered with an appropriate amount of PBS and irradiated in an ultraviolet crosslinker (SCIENTZ03-II, Ningbo Scientz Biotechnology Co., Ltd).”
“Absorbance was measured at 450 nm (Tecan M200 Infinite Pro, Tecan (Shanghai) Trading Co., Ltd.) to determine the IC50 of the UVA-stimulated cell dose. RP and RFP were dissolved in DMEM at 20.00, 10.00, 5.00, 2.50, 1.25, 0.63, and 0.31 g/L.”
“After full lysis and 12,000 g centrifugation ((Allegra X-30R, Beckman Coulter, Inc.) at 4℃ for 5 min, the supernatant was taken, and the total antioxidant capacity, MDA content and the activity of SOD, CAT, and GSH-Px were determined according to the kit instructions.”
“Then, the Shanghai Majorbio Bio-Pharm Technology Co., Ltd was commissioned to perform for high-throughput sequencing on an Illumina HiSeq 4000 sequencing plat-form.”
“With the adjusted P-value (P-adjust) < 0.05, | log2fold change | (| log2FC |) > 1.2 as the screening criteria, differential expression analysis was performed on the sequencing results using DESeq2 software. Goatools software was used for gene ontology (GO) enrichment analysis, and R script was used for the Kyoto Encyclopedia of Genes and Genomes (KEGG) pathway enrichment analysis. When P-adjust < 0.05, the GO function or KEGG Pathway was considered to be significantly enriched.
“According to the gene sequences published in NCBI, primers were designed by PrimerExpress software, and β-actin was used as the internal reference gene for RT-PCR validation (ABI7300, Thermo Fisher Scientific Co., Ltd.).”
Results
- This is not clear sentences. The average molecular weight of the amino acids was 128. The peptide sequencing 239 results show that RFP is a mixture of small peptides with less than 11 peptides, and the 240 contents of each component decrease from top to bottom. RFP is a mixture of small peptides with less than 11 amino acids?
Reply: Thanks for pointing out the question. We have reformulated the main idea of this sentence. RFP is a mixture of small peptides with less than 11 amino acids.
“The results of peptide sequencing are presented in Table 2. RFP is a mixture composed of seven small peptides, each of which has fewer than 11 amino acids. The mean molecular weight of amino acids is 128. The most abundant small peptide amino acid sequence is PLL, and the least abundant is YNEGDAPVVA.”
- It is not acceptable to publish experimental results and their presentation as scientific publications. The authors requested that the quality of the presentation of the data as well as the interpretation of the results be improved.
Reply: Thanks for your great suggestions which helped a lot for us to improve this manuscript. We have modified all the figures to present the data better and made the necessary changes to the interpretation of the results. The changes made to the manuscript were marked using the “Track Changes” function.
- The overall language of the manuscript is not well written.
Reply: Thanks for your comments. According to your advice, we have revised the unclear paragraphs in our manuscript, and our manuscript has been polished by the Beijing Wordstalk Co., Ltd.
- The name of the microorganism must be in italics.
Reply: Thanks for pointing out the question. This mistake has been corrected.
Reviewer 2 Report
After the review of manuscript "Peptides from Rice Fermentation of Lactobacillus plantarum protects UVA-induced Oxidative Stress in Skin", we found that RFP can alleviate oxidative stress caused by UVA. I believe this study is interesting and convincing. However, there are some problems should be well addressed.
1. In the case of RFPs composed of YNDGDAPIVA, YNDGDAPIV and YNEGDAPVVA, is novelty a guaranteed peptide?
2. 2. The difference between RP and RFP is that 11 smaller peptides are formed. Then, It is then recommended to present the specificities of the 11 small peptides in the results section.
3. In vitro experiment, it was confirmed that RFP has 10-30% more cytoprotective effect than RP. Nrf2, HO-1, NQO1, etc. were identified by ELISA as these molecular mechanisms. How did lysate prep proceed? Why did you use ELISA instead of western blot? Please specify the manufacturer and catalog number for the ELISA kit for the relevant protein.
4.It is written that VC was used as a positive control in in vivo GSH. What is VC, and why not use the same positive control?
5. By marking the arrows in the Figure 9 picture, you need to explain to the reader what the arrows represent.
6. The overall resolution of Figure is very low. The characters on the y-axis are not clearly visible.
7. The methods are generally very simple. Please indicate in detail about the fermentation process, purification process, and catalog number of the purchased product.
Author Response
Oct. 20, 2022
Manuscript ID: antioxidants-1970346
Type of manuscript: Article
Title: Purification and Identification of Antioxidant Peptides from Rice
Fermentation of Lactobacillus plantarum and their Protective Effects on
UVA-induced Oxidative Stress in Skin
Authors: Qiuting Mo, Shiquan You, Hao Fu, Dongdong Wang, Jiachan Zhang,
Changtao Wang, Meng Li *
Received: 29 September 2022
E-mails: 2130041028@st.btbu.edu.cn, shiquan.you@starealife.com, 20111006@th.btbu.edu.cn, wdd@btbu.edu.cn, 20120720@btbu.edu.cn, wangct@th.btbu.edu.cn, limeng@btbu.edu.cn
Dear referee,
Thank you very much for giving us this opportunity to revise our manuscript. The comments and suggestions made on our manuscript are very encouraging and helpful. Detailed point-by-point responses to your comments are provided in the following pages. Note that your comments are presented in italics, and our responses are in Roman and blue font. All changes made to the manuscript were marked using the “Track Changes” function. In addition, we addressed all these major points and other issues carefully and revised the manuscript accordingly. Please let me know if you have any further questions.
Sincerely,
Meng Li
Beijing Key Lab of Plant Resource Research and Development, College of Chemistry and Materials Engineering, Beijing Technology and Business University, Fucheng Road, Beijing 100048, China
Tel.: +86-13426015179
E-mail: limeng@btbu.edu.cn
- In the case of RFPs composed of YNDGDAPIVA, YNDGDAPIV and YNEGDAPVVA, is novelty a guaranteed peptide?
Reply: Thanks for giving us the opportunity to revise the manuscript and your comments.
It was found that RFP is a mixture of eight small peptides by peptide sequencing. YNDGDAPIVA, YNDGDAPIV and YNEGDAPVVA are three of them. When reviewing the literature, we did not find any literature related to YNDGDAPIVA, YNDGDAPIV, and YNEGDAPVVA. And through the references, we can speculate that YNDGDAPIVA, YNDGDAPIV, and YNEGDAPVVA may be one of the reasons for the antioxidant ability of RFP. Therefore, RFP is a mixed peptides with a novelty guarantee.
- The difference between RP and RFP is that 11 smaller peptides are formed. Then, it is then recommended to present the specificities of the 11 small peptides in the results section.
Reply: Thanks for your comments and constructive suggestions. In the results section, we added the specificity of the eight small peptides that make up the RFP.
“The results of peptide sequencing are presented in Table 2. RFP is a mixture composed of 8 small peptides, each of which has fewer than 11 amino acids. The mean molecular weight of amino acids is 128. The most abundant small peptide amino acid sequence is PLL, and the least abundant is YNEGDAPVVA. The amino acid composition of these small peptides is mostly hydrophobic amino acids (alanine (Ala,A), phenylalanine(Phe, F), isoleucine (Ile, I), leucine (Leu, L), proline (Pro, P), valine (Val, V)). The amino acid sequences of YNDGDAPIVA, YNDGDAPIV, FYNDGDAPIV and YNEGDAPVVA are relatively similar which except for hydrophobic amino acids, all contain tyrosine (Try,Y), glycine (Gly, G), asparagine (Asn, N) and aspartate (Asp, D). And all the tyrosine and acidic amino acid (D or glutamate (Glu, G)) are separated by an N. RP consists of 8 amino acids and the sequence is Lys-His-Asn-Arg-Gly-Asp-Glu-Phe. RP, YNDGDAPIVA, YNDGDAPIV, FYNDGDAPIV and YNEGDAPVVA, all contain N and G. RP and FYNDGDAPIV both contain F. Both RP and VRVF contain arginine (Arg, R). However, RP is mainly composed of basic amino acids (histidine (His, H), lysine (Lys, K) and R), and acidic amino acids (D, E).”
- In vitro experiment, it was confirmed that RFP has 10-30% more cytoprotective effect than RP. Nrf2, HO-1, NQO1, etc. were identified by ELISA as these molecular mechanisms. How did lysate prep proceed? Why did you use ELISA instead of western blot? Please specify the manufacturer and catalog number for the ELISA kit for the relevant protein.
Reply: Thanks for your comments. We have added lysate preparation in the methods section. We have supplemented the manufacturer and catalog number for the ELISA kit for the relevant protein in the materials section and added lysate preparation in the methods section. Compared with western blot, ELISA can specifically and quantitatively detect the protein contents of NRF2, NQO1 and HO-1, and observe the influence of NRF2, NQO1 and HO-1 before and after stimulation, as well as before and after sample action. Moreover, ELISA is more straightforward and faster in operation. Therefore, we finally chose to perform ELISA instead of western blot.
“Human Nuclear Factor E2-related Factor 2 (NRF2) ELISA Kit, Human Quinone NADH Dehydrogenase 1 (NQO1) ELISA Kit and Human Heme Oxygenase-1 (HO-1) ELISA Kit were purchased from Cusabio Biotech Co., Ltd (Houston, USA).”
“HSF were laid in corning 25cm2 rectangular created neck cell culture flask with vent caps at a density of 5 × 106 cells/ flask for 12 h. After sample treatment and UVA irradiation, nuclear and cytoplasmic proteins were prepared according to the nuclear and cytoplasmic protein extraction kit. Then they were used to determine the NRF2 Protein content using Human Nuclear Factor E2-related Factor 2 (NRF2) ELISA Kit.
HSF were laid in corning 75cm2 rectangular created neck cell culture flask with vent caps at a density of 1.5 × 107 cells/ flask for 12 h. After sample treatment and UVA irradiation, the preparation of cell lysates and the detection of protein contents of NQO1 and HO-1 were performed according to Human Quinone NADH Dehydrogenase 1 (NQO1) ELISA Kit and Human Heme Oxygenase-1 (HO-1) ELISA Kit.
Specific test protocols refer to the kit instructions.”
- It is written that VC was used as a positive control in in vivo GSH. What is VC, and why not use the same positive control?
Reply: Thanks for pointing out the questions. VC is vitamin C. Since our fermentation product RFP was a mixture of small peptides, GSH was selected as a positive control in both cell and animal experiments because GSH is a known peptide antioxidant. Anyway, I'm sorry for the trouble caused by miswriting GSH as VC in the animal part.
- By marking the arrows in the Figure 9 picture, you need to explain to the reader what the arrows represent.
Reply: Thanks for your great suggestions which helped a lot for us to improve this manuscript. We've annotated the arrows in Figure 9 and explained what they mean.
“Single arrow indicates irregular epidermal hyperplasia; the double arrow points to the corner plug of the hair follicle.”
- The overall resolution of Figure is very low. The characters on the y-axis are not clearly visible.
Reply: Thanks for your comments. We have modified all the figures to present the data better.
- The methods are generally very simple. Please indicate in detail about the fermentation process, purification process, and catalog number of the purchased product.
Reply: Thanks for your comments. We have added or modified the problems you pointed out.
“The MRS medium was configured and sterilized by autoclaving for 15min. After cooling to room temperature, Lactobacillus suspension was added for expansion culture. The culture conditions were 37℃ and 180rpm for 24h.”
“Deionized water was added at a material-to-liquid ratio of 1:4. After sterilization, 5% activated Lactobacillus plantarum liquid (107 CFU/mL) was added, and fermentation was performed at 180 r/min at 37℃.”
“After centrifugation, the polysaccharides of the supernatant were removed by the alcohol precipitation method and the ratio of ethanol to supernatant was 4:1. After freeze-drying at -60℃ in vacuum rotation, the freeze-dried rice fermentation liquid powder was obtained and the yield was 0.17 g/L.”
“The sample loading volume was 2 mL and the flow rate was 2 mL/min. They were then eluted sequentially with deionized water, 0.1 NaCl solution, and 1 mol/L NaCl solution at a flow rate of 2 mL/min.”
“After being blown evenly, HSF were immediately transferred to a culture flask containing an appropriate amount of medium and placed in a 5% CO2 incubator (Thermo Fisher Scientific Co., Ltd.) at 37℃ for resuscitation.”
“HSF were inoculated in 96-well plates at a density of 8 × 103 cells/well for 12 h, then covered with an appropriate amount of PBS and irradiated in an ultraviolet crosslinker (SCIENTZ03-II, Ningbo Scientz Biotechnology Co., Ltd).”
“Absorbance was measured at 450 nm (Tecan M200 Infinite Pro, Tecan (Shanghai) Trading Co., Ltd.) to determine the IC50 of the UVA-stimulated cell dose. RP and RFP were dissolved in DMEM at 20.00, 10.00, 5.00, 2.50, 1.25, 0.63, and 0.31 g/L.”
“After full lysis and 12,000 g centrifugation ((Allegra X-30R, Beckman Coulter, Inc.) at 4℃ for 5 min, the supernatant was taken, and the total antioxidant capacity, MDA content and the activity of SOD, CAT, and GSH-Px were determined according to the kit instructions.”
“Then, the Shanghai Majorbio Bio-Pharm Technology Co., Ltd was commissioned to perform for high-throughput sequencing on an Illumina HiSeq 4000 sequencing plat-form.”
“With the adjusted P-value (P-adjust) < 0.05, | log2fold change | (| log2FC |) > 1.2 as the screening criteria, differential expression analysis was performed on the sequencing results using DESeq2 software. Goatools software was used for gene ontology (GO) enrichment analysis, and R script was used for the Kyoto Encyclopedia of Genes and Genomes (KEGG) pathway enrichment analysis. When P-adjust < 0.05, the GO function or KEGG Pathway was considered to be significantly enriched.
“According to the gene sequences published in NCBI, primers were designed by PrimerExpress software, and β-actin was used as the internal reference gene for RT-PCR validation (ABI7300, Thermo Fisher Scientific Co., Ltd.).”
Round 2
Reviewer 2 Report
The authors responded appropriately to the reviewer's comments.
Author Response
Detailed response to reviewer 2’s comments
Nov. 13, 2022
Manuscript ID: antioxidants-1970346
Type of manuscript: Article
Title: Purification and Identification of Antioxidant Peptides from Rice Fermentation of Lactobacillus plantarum and their Protective Effects on UVA-induced Oxidative Stress in Skin
Authors: Qiuting Mo, Shiquan You, Hao Fu, Dongdong Wang, Jiachan Zhang, Changtao Wang, Meng Li *
Received: 29 September 2022
E-mails: 2130041028@st.btbu.edu.cn, 13269182262@163.com, 18811359600@163.com, wdd@btbu.edu.cn, xiaochan8787@163.com, wangct@th.btbu.edu.cn, limeng@btbu.edu.cn
Dear referee,
We are appreciated for your time on reviewing the manuscript. The comments and suggestions made on our manuscript are very encouraging and helpful.
Sincerely,
Meng Li
Beijing Key Lab of Plant Resource Research and Development, College of Chemistry and Materials Engineering, Beijing Technology and Business University, Fucheng Road, Beijing 100048, China
Tel.: +86-13426015179
E-mail: limeng@btbu.edu.cn